# Does the Pathologic Fracture Predict Severe Paralysis in Patients with Metastatic Epidural Spinal Cord Compression (MESCC)?—A Retrospective, Single-Center Cohort Analysis

**DOI:** 10.3390/jcm12031167

**Published:** 2023-02-01

**Authors:** Lukas Klein, Georg W. Herget, Gabriele Ihorst, Gernot Lang, Hagen Schmal, Ulrich Hubbe

**Affiliations:** 1Department of Orthopedics and Trauma Surgery, Faculty of Medicine, Medical Center—University of Freiburg, 79106 Freiburg, Germany; 2Comprehensive Cancer Center Freiburg (CCCF), Faculty of Medicine, Medical Center—University of Freiburg, 79106 Freiburg, Germany; 3Clinical Trials Unit, Faculty of Medicine, Medical Center—University of Freiburg, 79106 Freiburg, Germany; 4Department of Neurosurgery, Faculty of Medicine, Medical Center—University of Freiburg, 79106 Freiburg, Germany

**Keywords:** spinal metastases, vertebral fracture, metastatic epidural spinal cord compression (MESCC), paralysis, decompression, stabilization, recovery

## Abstract

Currently, there is uncertainty about the predictive factors for metastatic epidural spinal cord compression (MESCC) and consecutive symptomatology in tumor patients. Prognostic algorithms for identifying patients at risk for paralysis are missing. The influence of the pathologic fracture on the patient’s symptoms is widely discussed in the literature and we hypothesize that pathologic fractures contribute to spinal cord compression and are therefore predictive of severe paralysis. We tested this hypothesis in 136 patients who underwent surgery for spinal metastases. The most common primary cancers were prostate (24.3%, n = 33), breast (11.0%, n = 15), lung (10.3%, n = 14), and cancer of unknown primary (10.3%, n = 14). MESCC primarily affected the thoracic (77.2%, n = 105), followed by the lumbar (13.2%, n = 18) and cervical (9.6%, n = 13) spine. Pathologic fractures occurred in 63.2% (n = 86) of patients, mainly in osteolytic metastases. On the American spinal injury association (ASIA) impairment scale (AIS), 63.2% (n = 86) of patients exhibited AIS grade D and 36.8% (n = 50) AIS grade C-A preoperatively. The presence of a pathologic fracture alone did not predict severe paralysis (AIS C-A, *p* = 0.583). However, the duration of sensorimotor impairments, patient age, spinal instability neoplastic score (SINS), and the epidural spinal cord compression (ESCC) grade together predicted severe paralysis (*p* = 0.006) as did the ESCC grade 3 alone (*p* = 0.028). This is in contrast to previous studies that stated no correlation between the degree of spinal cord compression and the severity of neurologic impairments. Furthermore, the high percentage of pathologic fractures found in this study is above previously reported incidences. The risk factors identified can help to predict the development of paralysis and assist in the improvement of follow-up algorithms and the timing of therapeutic interventions.

## 1. Introduction

With the increasing survival of cancer patients, the treatment of skeletal and especially spinal metastases has become more important as they can cause a myriad of skeletal-related events (SRE), including hypercalcemia, pathologic fractures, spinal cord compression, and the necessity for radiation or surgery [1,2,3]. Metastatic epidural spinal cord compression (MESCC) affects 5–14% of all cancer patients, and subsequent sensorimotor impairments significantly lower their quality of life and life expectancy [2,4,5]. In most cases, vertebral metastases do not infiltrate the dura mater, but cause damage to the spinal cord through compression [3]. Additional pathologic vertebral fractures, mostly compression fractures, may contribute independently to the development of subsequent sensorimotor impairments by causing spinal instability or the extrusion of the metastasis to the spinal canal (Figure 1) [6,7].

Besides the patient’s history, clinical examination and imaging, such as computed tomography (CT) and magnetic resonance imaging (MRI), are tools to detect spinal metastases and estimate the extent of MESCC as well as the risk or presence of vertebral fractures [8]. Those are the decision-making aids to guide the timing of therapeutic interventions, such as surgery, as well as the surgical technique [9,10].

Currently, we do not know in detail which patients will develop MESCC and consecutive neurologic impairments, such as paralysis. A number of predictive factors have been tested but even a National Institute for Health Research (NHS) review states that there is uncertainty about their potential and that clear prognostic algorithms are missing [4].

Vertebral compression fractures have been shown to negatively affect the postoperative outcome of patients with MESCC. We hypothesize that pathologic fractures contribute to spinal cord compression by extrusion of the vertebral metastasis to the spinal canal and therefore cause more severe impairments in patients with MESCC [9,10].

## 2. Materials and Methods

### 2.1. Data Source and Patient Population

A database was generated of 136 patients operated on for paralysis following MESCC at the Department of Neurosurgery and the Department of Orthopedics and Trauma surgery at the Comprehensive Cancer Center Freiburg (CCCF) between 2012 and 2019 (Figure 2). The exclusion criteria were previous surgery at the affected spinal level, intracerebral metastasis, invasive primary tumor, and invasive paravertebral metastasis.

### 2.2. Data Collection

Patient age at the time of surgery, sex, physical status according to the American Society of Anesthesiologists (ASA) classification system, and the duration of symptoms were obtained from digital medical records [11]. The primary cancer was identified by histologic examination, and osteolyses caused by multiple myeloma were subsumed under metastases. CT and MRI examinations conducted immediately prior to surgery were reviewed, and we assessed the location of the most severe spinal cord compression, metastatic growth pattern (i.e., osteolytic, osteoplastic, or mixed), grade of epidural spinal cord compression (ESCC scale), the spinal instability neoplastic score (SINS), Taneichi score (in osteolytic metastases), and the fractured vertebra’s loss of height [12,13,14].

A neurologist or neurosurgeon assessed the patient’s neurological status at the time of admission and discharge, documenting pain, sensory, motor deficits, and bowel and bladder dysfunction. Paralysis was classified relying on the American Spinal Injury association’s (ASIA) impairment scale (AIS) [15]. To assess the influence of potential contributing factors on the severity of sensorimotor impairments, patients were divided into two groups with preoperatively severe (AIS A-C) versus moderate (AIS D) impairments.

### 2.3. Statistical Analysis

We conducted statistical analyses using SPSS 27.0 (IBM Corp., Armonk, NY, USA) and SAS 9.2 (SAS Institute Inc., Cary, NC, USA). Comparisons were made with Fisher’s exact test for dichotomous variables. To identify parameters potentially predicting severe impairments (AIS A-C), we conducted multivariate logistic regression analyses. Parameters showing univariate association with motor impairment severeness were included as independent variables: patient age (<61, 61–67, 68–76, and >77 years, respecting this cohort’s distribution), cancer type, ESCC (incomplete = grade 1–2, complete = grade 3), paralysis duration (in days), and SINS-values (<8, 8–9, 10–12, and >13) were combined into a model. Fractures were not included, as they are strongly collinear with the SINS, and the latter correlated linearly with the consequent height loss. Backward variable selection was performed (selection criterion *p* < 0.2) to obtain the final model, and the cancer type was eliminated. Results were considered statistically significant if the *p*-value was <0.05.

## 3. Results

### 3.1. Patients’ Characteristics and Primary Tumors

A total of 136 patients were included of whom 66.2% were male (n = 90) and 33.8% female (n = 46). The mean age of patients at the time of surgery was 67.2 years (range 21 to 89 years) and differed slightly between males (67.5 years) and females (66.5 years). Most patients were ASA 3 (62.5%, n = 85), followed by 4 (16.9%, n = 23) and 2 (12.5%, n = 17). The duration of motoric impairments before surgery was documented in 125 patients and was a mean of 7.38 days (range 0–42). This differed substantially between patients with minor and severe motor impairments (Table 1).

Prostate cancer (24.3%, n = 33), breast cancer (11.0%, n = 15), cancer of unknown primary (10.3%, n = 14), lung cancer (10.3%, n = 14), multiple myeloma (5.9%, n = 8), and renal cell cancer (5.9%, n = 8) were the most common primary cancers (Table 2).

### 3.2. Symptoms

Before surgery, 63.2% were classified AIS D (n = 86), 25.0% AIS C (n = 34), 5.9% AIS B (n = 8) and 5.9% AIS A (n = 8). Concomitant symptoms were present in most patients. A total of 66.7% suffered from pain at the affected level of the spine (n = 90/135), 71.6% suffered sensory deficits (n = 62/134) and 45.6% had bladder and/-or rectal dysfunction (n = 62/136). A total of 8.2% (n = 11/134) of patients suffered from neither sensory impairments nor pain. One patient’s pain documentation was insufficient and sensory function was not documented in two patients. Paralysis was the initial cancer manifestation in 27.9% (n = 38).

### 3.3. Metastatic Profile

Metastases affected primarily the thoracic, followed by the lumbar and cervical spine (Table 3). The radiographic growth pattern assessed using CT, was usually osteolytic (67.4%, n = 91) and less frequently osteoplastic (17.8%, n = 24) or mixed (14.8%, n = 20, Table 3). Metastatic growth was not assessable in one patient because of complete vertebral destruction. Pathologic fractures occurred in conjunction with 63.2% (n = 86) of all metastases, and osteolytic metastases exhibited a higher prevalence (76.9%, n = 70) than mixed (55.0%, n = 11) or solely osteoplastic metastases (20.8%, n = 5). An osteolytic growth pattern significantly predicted the presence of a pathologic fracture (*p* < 0.001).

The SINS was determined in 132 of 136 lesions; it amounted to a mean of 10.25 points (range 3–17) and revealed a linear correlation with the presence of pathologic fractures. The observed height loss was up to 25% in 35 (mean SINS 10.68), 26–50% in 20 (mean SINS 11.7), 51–75% in 18 (mean SINS 13), and 76–100% in 13 (mean SINS 14.17) patients.

For osteolytic metastases of the thoracic and lumbar spine, the Taneichi score was determined in n = 87/136. The most common growth pattern at the thoracic level was Taneichi F (n = 43), followed by A (n = 8), B (n = 7), D (n = 7), E (n = 4), and C (n = 2). At the lumbar level, Taneichi E (n = 6) was more common than Taneichi D (n = 4), C (n = 4) and F (n = 2).

The grade of spinal cord compression according to the ESCC scale was evaluated in 135 patients (the preoperative MRI of one patient was missing). ESCC was grade 3 in 56.3% (n = 76), grade 2 in 42.2% (n = 57), and grade 1b in 1.5% (n = 2) of patients. No grades 0, 1a, or 1c were observed (Table 3).

### 3.4. Sensorimotor Impairments

Severe impairments were detected in 36.8% (n = 50), and moderate in 63.2% (n = 86) of patients. The presence of a pathologic fracture alone did not predict severe paralysis, while a complete spinal cord compression, defined as ESCC grade 3, significantly predicted severe impairments (Table 4).

To investigate the influence of other contributing factors, patient age, complete spinal cord compression (ESCC grade 3), duration of paralysis, and SINS values were tested using multivariate logistic regression. These factors together were highly predictive of severe sensorimotor impairments (*p* = 0.006, Table 5).

### 3.5. Surgical Approach, Early Neurological Outcome and Complications

Patients with severe impairments underwent surgery 3.11 days sooner after their paralysis onset than patients with moderate impairments. A total of 34.4% (n = 43/125) were operated on within the first 48 h after the onset of symptoms. The duration of paralysis was not documented in 11 patients. Metastases were treated with one surgery in 91.9% (n = 125) and with two in 8.1% of patients (n = 11). The mean duration of inpatient treatment at our hospital was 7.3 days (range 4–41 days).

At the time of discharge, the sensorimotor performance was AIS D in 66.7% (n = 90), AIS C in 17.0% (n = 23), AIS E in 7.4% (n = 10), AIS A in 5.2% (n = 7), and AIS B in 3.7% (n = 5) of patients. A total of 69.1% (n = 94) presented the same neurological status as they had preoperatively. The neurologic status of 25.0% (n = 34) improved at least by one grade on the AIS, and 5.1% (n = 7) suffered from deterioration. One patient’s postoperative score could not be determined because of his sudden death.

Perioperative complications, such as pneumonia, urinary tract infection, wound infection, or postoperative delirium occurred in 19.85% (n = 27); 4.41% (n = 6) of patients died perioperatively.

## 4. Discussion

This study examined the relevance of pathologic fractures in patients with MESCC and paralysis as well as their predictive value for severe paralysis. Compared to previous studies, pathologic fractures were more common and were detected in over 60% of patients. However, their presence did not predict more severe symptoms. Complete spinal cord compression alone, combined with patient age, paralysis duration and SINS proved to be predictive of severe sensorimotor impairments. These should be considered in follow-up algorithms of tumor patients to detect patients at high risk for paralysis.

### 4.1. Epidemiology

Our cohort represents the distribution of cancer in the general population and their likelihood to generate bone metastases. Our results are in accordance with German, European, and American cancer registers which show that men are more likely to suffer from cancer than women [16,17,18]. Prostate cancer is the most common malignancy in patients with bone metastases, followed by breast and lung cancer [18].

### 4.2. Symptoms

In the 1980s and 1990s, about 20% of patients presented with spinal epidural metastasis as their initial manifestation of cancer [19]. In our study, this percentage was 27.9%, even though most patients exhibited concomitant symptoms, known as “red flags” or “warning signs”. In our cohort, less than 10% of patients suffered from neither sensory impairments nor pain. Previous findings support the observation that pain and sensory loss are very frequent, and that bladder and sphincter dysfunction coincide with more advanced cancers involving spinal metastases [20]. The awareness on the part of physicians and patients is, therefore, crucial to ensure a timely diagnosis and consequent treatment [21].

Another third of our patients developed sensorimotor impairments within the first three months after their tumor diagnosis. This is striking, as MESCC occurs in association with advanced cancer dissemination, but there is also evidence that in up to 23.4% of patients, bone metastases appear during the first three months after their cancer diagnosis [1,18]. Initial interpretation of bone scans by an interdisciplinary tumor board, as well as follow-up algorithms, should take these observations into account.

### 4.3. Metastatic Profile

The thoracic spine is reportedly affected in 60% of patients with MESCC, followed by the lumbar in 25% and cervical in 15% [2]. We identified slightly more (77.2%) thoracic metastases in our population, but the distribution pattern resembled those in prior investigations [4,7].

According to previous studies, the incidence of pathologic fractures in patients diagnosed with MESCC is 30–40% [7]. We observed about twice as many fractures, and their prevalence in this study was 63.2%. This might be attributable to two facts: First, patients who underwent prophylactic stabilization were excluded from our analysis, and we therefore might have assessed less unstable vertebrae. Second, in about a quarter of our patients, paralysis was the first symptom of a cancer disease, which might be because their metastases were rapidly growing.

Fractures were not only associated with osteolytic and mixed metastases. Over one-fifth of osteoplastic metastases, often regarded as “stable”, revealed pathologic fractures.

### 4.4. Pathogenesis of Sensorimotor Impairments

Paralysis in patients suffering from MESCC is triggered by direct pressure on the spinal cord [2]. Various factors can contribute to the severity of impairments, such as long-lasting pressure, complete obstruction of the spinal canal and/or pathologic fractures (Figure 3).

Our hypothesis that vertebral compression fractures cause more severe paralysis in patients with MESCC was refuted. On the other hand, complete spinal cord compression was predictive of advanced paralysis in this cohort. Previously, Uei et al. detected no correlation between the degree of spinal cord compression in MRI studies and paralysis severity in 467 patients [22]. This effect is potentially attributable to our decision to only discriminate between complete and incomplete spinal cord compression, but it should be taken into account when consulting patients with high-grade spinal cord compression. In addition, we found that age, complete spinal cord compression (ESCC grade 3), SINS values, and paralysis duration were predictive of severe impairments in patients with MESCC. We propose these parameters be implemented in prognostic algorithms to predict paralysis in patients with MESCC and to be tested in prospective studies.

### 4.5. Surgical Approach, Complications, and Early Postoperative Outcome

There has long been controversy about what constitutes optimal MESCC treatment, and additional surgery is sometimes regarded as not beneficial or even inferior to radiation alone [23,24]. A systematic Cochrane review on MESCC treatment argued that surgery should be restricted to younger patients with paraplegia lasting less than 48 h [24]. A small percentage of our cohort underwent surgery within this brief time frame, and the mean latency was close to a week. This came about because of a delayed external diagnosis. Some patients had not seen a physician at all beforehand, and in other cases, the treating physicians failed to refer the patient to a specialized center in time. Patients with severe impairments underwent surgery nearly three days sooner after their paralysis onset than patients with moderate symptoms (5.46 vs. 8.57 d). The risk acceptance of patients and surgeons might differ in such dire situations.

Although most of our patients did not fulfill the aforementioned criteria, 25.0% presented improved sensorimotor function, with 7.4% recovering completely after their surgery. However, regaining function after MESCC is likely to take much longer than the average 7.3-day stay at our hospital [25].

Spinal cord relief can be achieved through posterior decompression via laminectomy alone, but our observation that 63.2% of metastases also revealed pathologic fractures emphasizes the need for stabilization in these patients. This claim is supported by the latest findings showing that additional spinal fusion, compared to decompression alone, reduces the patient’s risk of being readmitted within 30 days after surgery [26].

The rate of perioperative progression of symptoms, complications, and deaths we observed highlights the importance of a thorough risk evaluation to do no further harm. Scores such as the Tokuhashi, Tomita, Baur, Linden, Rades, or Katagiri scores have been established to help clinicians evaluate their patients’ prognoses, and they should be implemented during decision-making [27,28,29,30,31,32].

### 4.6. Limitations

This study examines a large cohort of patients with MESCC who underwent surgery for paralysis. We investigated the profile of spinal metastases and its potential influence on the severity of impairments. The accuracy of recorded symptoms was very high thanks to thorough documentation. The metastatic profile being reported is one of the strengths of this study, as it is presented in great detail and takes into account all relevant factors, including growth pattern, SINS, Taneichi score, ESCC grade, and the degree of vertebral body compression.

However, as this study was planned as a retrospective analysis, there is a selection bias as only operated patients were included. Patients with MESCC who underwent urgent radiation or were treated palliatively were not examined. We can only report on their acute postoperative outcomes, as most patients went to rehabilitation facilities after surgery and were not regularly seen for follow-up at our institution.

## 5. Conclusions

Pathologic vertebral fractures are more common in patients with MESCC than previously described. However, they do not predict severe paralysis on their own. In contrast, complete spinal cord compression (ESCC grade 3) was found to be a significant predictor of severe impairments. Further prognostic factors identified were patient age, duration of paralysis, and SINS values. These should be implemented in future follow-up algorithms, to help patients avoid neurological complications. The postoperative results demonstrated in this study show that surgical concepts relying on stabilization and decompression are capable to reverse paralysis already in the acute recovery stage. Due to the complex decision-making, cases should always be discussed by tumor boards.

## Figures and Tables

**Figure 1 jcm-12-01167-f001:**
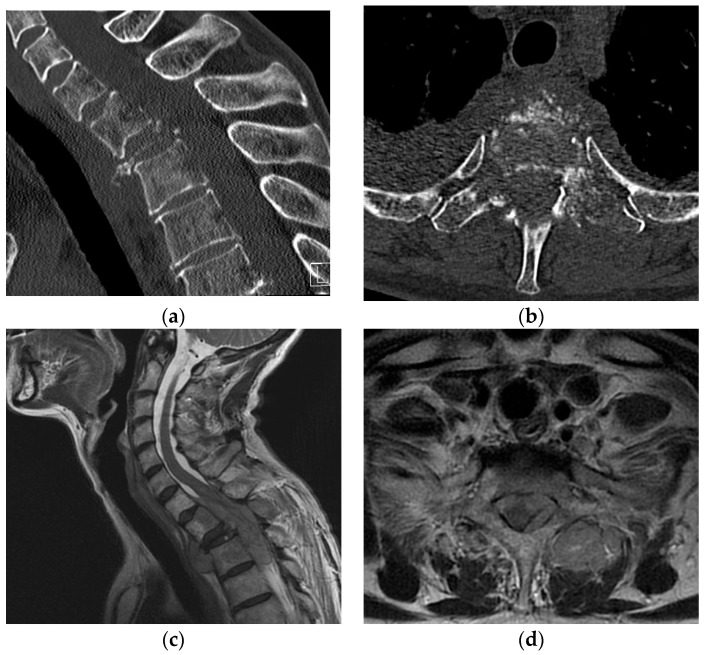
(**a**–**d**): Computed tomography in the sagittal (**a**) and transversal (**b**) plane of the cervical and upper thoracic spine of a 61-year-old female patient with breast cancer, who presented with progredient paralysis and gait disorder at our hospital. (**c**) Sagittal and (**d**) transversal view of T2-weighted magnetic resonance images showing complete spinal cord compression: ESCC grade 3 with no cerebrospinal fluid around the cord. ESCC, epidural spinal cord compression.

**Figure 2 jcm-12-01167-f002:**
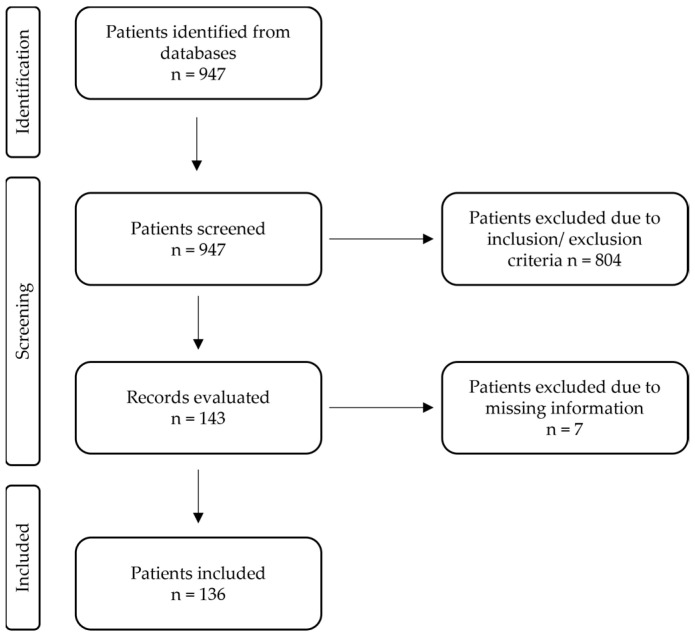
Patient selection.

**Figure 3 jcm-12-01167-f003:**
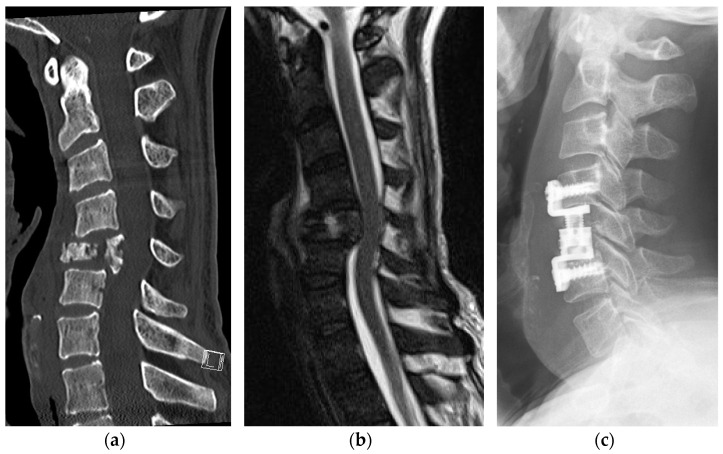
(**a**–**c**): Computed tomography in the sagittal plane (**a**) and sagittal view of T2-weighted magnetic resonance images (**b**) of a 48-year-old male patient with renal cell cancer. The pathologic fracture of the fifth cervical vertebra caused spinal instability and posterior dislocation of the vertebral body with incomplete spinal cord compression (ESCC grade 2). The lateral radiograph (**c**) demonstrates the postoperative result following corpectomy and cage implantation.

**Table 1 jcm-12-01167-t001:** Patients’ characteristics.

	Total n = 136	Minor Impairments (AIS D) n = 86	Severe Impairments (AIS A-C) n = 50
**Sex**			
male	90 (66.2%)	53 (61.6%)	37 (74.0%)
female	46 (33.8%)	33 (38.4%)	13 (26.0%)
**Age at surgery, y**			
median (range)	67.2 (21–89)	66.7 (21–68)	68.0 (30–84)
**ASA**			
ASA 1	0 (0%)	0 (0%)	0 (0%)
ASA 2			
ASA 3	17 (12.5%)	12 (14.0%)	5 (10.0%)
ASA 4	85 (62.5%)	55 (64.0%)	30 (60.0%)
missing	23 (16.9%)	13 (15.1%)	10 (20.0%)
	11 (8.1%)	6 (7.0%)	5 (10.0%)
**Duration of** **impairments, d**			
median (range)	7.38 (0–42)	8.57 (0–42)	5.46 (0–42)

Patient characteristics: age at the time of surgery, sex, ASA classification, duration of impairments. y, years; d, days; ASA, American society of anesthesiologists; AIS, American spinal injury association (ASIA) impairment scale.

**Table 2 jcm-12-01167-t002:** Cancer type.

Cancer Type	n = 136 (100%)
Prostate cancer	33 (24.3%)
Breast cancer	15 (11.0%)
Cancer of unknown primary (CUP)	14 (10.3%)
Lung cancer	14 (10.3%)
- Non small cellular lung cancer (NSCLC)	11 (8.1%)
- Small cellular lung cancer (SCLC)	3 (2.2%)
Multiple myeloma	8 (5.9%)
Renal cell cancer	8 (5.9%)
Skin cancer	5 (3.7%)
Lymphomas	5 (3.7%)
Colorectal cancer	5 (3.7%)
Hepatocellular carcinoma (HCC)	4 (2.9%)
Esophageal cancer	4 (2.9%)
Cancer of the female reproductive system	4 (2.9%)
Primary bone cancer	3 (2.2%)
Oropharyngeal cancer	3 (2.2%)
Bladder cancer	2 (1.5%)
Thyroid carcinoma	2 (1.5%)
Others	7 (5.1%)

60.0% of patients (n = 78) presented isolated bone metastases and 40.0% (n = 52) suffered from additional visceral metastases. Staging documentation was incomplete in six patients.

**Table 3 jcm-12-01167-t003:** Metastatic profile.

	All Metastases	Fracture	No Fracture
**Localization**	n = 136	n = 86	n = 50
cervical			
thoracic	13 (9.6%)	11 (12.8%)	2 (4%)
lumbar	105 (77.2%)	63 (73.3%)	42 (84%)
	18 (13.2%)	12 (14.0%)	6 (12%)
**Metastatic growth**	n = 135 (100%) *	n = 86 (63.7)	n = 49 (36.3%)
osteolytic	91 (67.4%)	70 (76.9%)	21 (23.1%)
osteoplastic	24 (17.8%)	5 (20.8%)	19 (79.2%)
mixed	20 (14.8%)	11 (55.0%)	9 (45.0%)
**SINS**	n = 132 (100%) *	n = 84 (63.6%)	n = 48 (36.4%)
median SINS (range)	10.25 (3–17)	11.92 (5–17)	7.33 (3–14)
**Taneichi Score**	n = 87 (100%)	n = 60 (69.0%)	n = 27 (31.0%)
thoracic A	8 (9.2%)	3 (5.0%)	5 (18.5%)
lumbar A	0 (0%)	0 (0%)	0 (0%)
thoracic B	7 (8.0%)	5 (8.3%)	2 (7.4%)
lumbar B	0 (0%)	0 (0%)	0 (0%)
thoracic C	2 (2.3%)	2 (3.3%)	0 (0%)
lumbar C	4 (4.6%)	2 (3.3%)	2 (7.4%)
thoracic D	7 (8.0%)	7 (11.7%)	0 (0%)
lumbar D	4 (4.6%)	3 (5.0%)	1 (3.7%)
thoracic E	4 (4.6%)	1 (1.7%)	3 (11.1%)
lumbar E	6 (6.9%)	6 (10.0%)	0 (0%)
thoracic F	43 (49.4%)	30 (50.0%)	13 (48.1%)
lumbar F	2 (2.3%)	1 (1.7%)	1 (3.7%)
lumbar G	0 (0%)	0 (0%)	0 (0%)
**ESCC grade**	n = 135 (100%) *	n = 85 (63.0%)	n = 50 (37.0%)
1b	2 (1.5%)	1 (1.2%)	1 (2.0%)
2	57 (42.2%)	39 (45.9%)	18 (36.0%)
3	76 (56.3%)	45 (52.9%)	31 (62.0%)

Metastatic profile with spinal localization, metastatic growth pattern, SINS, Taneichi score, and initial ESCC grade for fractured and non-fractured vertebrae. SINS, spine instability neoplastic score; ESCC, epidural spinal cord compression. * = missing values: In one case metastatic growth pattern could not be identified due to the destruction of the vertebra; in four cases the SINS could not be determined due to missing information; in one case the ESCC grade could not be measured due to a missing MRI study.

**Table 4 jcm-12-01167-t004:** Descriptive analysis.

	Moderate Impairments AIS D (n = 86)	Severe Impairments	*p*-Value
		AIS A, B, C (n = 50)	
ESCC 1b and 2 (n = 59)	50.6% (n = 43)	32.0% (n = 16)	
ESCC 3 (n = 76)	49.4% (n = 42)	68.0% (n = 34)	0.0479
No fracture (n = 50)	34.9% (n = 30)	40.0% (n = 20)	
Fracture (n = 86)	65.1% (n = 56)	60.0% (n = 30)	0.5835

Descriptive analysis for predicting severe neurologic impairments with the presence of complete spinal cord compression (ESCC-scale grade 3) versus incomplete spinal cord compression (ESCC-scale grade 1b, 2) and the presence of a fracture using Fisher’s exact test. ESCC, epidural spinal cord compression; AIS, American spinal injury association (ASIA) impairment scale.

**Table 5 jcm-12-01167-t005:** Multivariate logistic regression analysis.

	Odds Ratio	95% CI	*p*-Value
**Duration of paralysis, d**			
continuous (range 0–42)	0.95	0.899–1.006	0.07
**Age, y**	2.201	0.648–7.469	
61–67 vs. <61	4.408	1.319–14.737	0.12
68–76 vs. <61	2.624	0.786–8.759	
>76 vs. <61			
**ESCC**	2.582	1.104–6.038	0.02
3 vs. 1b and 2			
**SINS**			
8–9 vs. <8	2.422	0.752–7.803	
10–12 vs. <8	1.892	0.613–5.833	0.12
13–18 vs. <8	0.661	0.199–2.196	

Multivariate logistic regression analysis for predicting severe neurologic impairments. ASIA, American spinal injury association impairment scale; CI, confidence interval; y, years; ESCC, epidural spinal cord compression; SINS, spinal instability neoplastic score.

## Data Availability

The data presented in this study are available on request from the corresponding author. The data are not publicly available due to the detailed analysis which comprises the dates of inpatient treatment, diagnosis, and complications. With these data, it is possible to draw conclusions about individual patient histories. German data protection regulations put these data under strict control.

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
