# Peer review of "Does the Pathologic Fracture Predict Severe Paralysis in Patients with Metastatic Epidural Spinal Cord Compression (MESCC)?—A Retrospective, Single-Center Cohort Analysis"

_jcm, 2023, doi:10.3390/jcm12031167_

Round 1
Reviewer 1 Report
The article is interesting, but it needs some corrections.
Please clearly associate the title of the article with the purpose of the research. Describe your research project. Provide a hypothesis and research questions.
Give the strengths and weaknesses of the presented research in the Discussion.
What is the practical application of research results.
Conclusions should be related to the research objective and research questions.
Author Response
Dear reviewer,
thank you very much for the helpful comments on our work. We appreciate all of them and think that the revisions really made us help improve the coherence of the manuscript.
- Title: We changed the title according to the main research question: Do pathologic fractures predict severe paralysis in patients with MESCC? This Furthermore we added the primary hypothesis and the research questions to the abstract and the introduction.
- Strength and weaknesses: Were further clarified in the discussion as well as in the conclusion.
- Practical application: The practical apllication is to implement the predictive factors found in this work into follow up algorithms for tumor patients with spinal metastases. This was added to the discussion and conclusion. We also prepare for the practical apllication already in the introduction when citing the NHS review and the absence of prognostic algorithms.
- Conclusions: Were specified and some less related statements were removed.
We are looking forward to your response, yours faithfully,
Lukas Klein
Reviewer 2 Report
The authors propose an original article on MESCC. The topic is interesting, however the purpose of the study is not clear; beyond describing their experience, authors should give a better overview of the background and current literature, to identify one or more precise study questions.
Otherwise, the article is well organized, with good methodology and clear presentations of results.
More images are needed, also including post surgical results.
Minor grammar and spelling errors through the manuscript.
Author Response
Dear Reviewer,
thank you very much for the helpful comments on our work. We appreciate all of them and think that the revisions really made us help improve the coherence of the manuscript.
- Overview of background and research question: We changed the title according to the main research question: Do pathologic fractures predict severe paralysis in patients with MESCC? Furthermore we added the primary hypothesis and the research questions to the abstract and the introduction. The introduction was revised and the research background is better described, citing the latest NHS review. This helps understand the lack of prognostic factors and algorithms in these patients.
- Images: A second case was added of a patient with a cervical metastasis demonstrating the preoperative condition and the postoperative results.
Further changes were made in the discussion and Conclusion according to your colleague's review.
We are looking forward to your response, yours faithfully,
Lukas Klein